# In Search of Species-Specific SNPs in a Non-Model Animal (European Bison (*Bison bonasus*))—Comparison of De Novo and Reference-Based Integrated Pipeline of STACKS Using Genotyping-by-Sequencing (GBS) Data

**DOI:** 10.3390/ani11082226

**Published:** 2021-07-28

**Authors:** Sazia Kunvar, Sylwia Czarnomska, Cino Pertoldi, Małgorzata Tokarska

**Affiliations:** 1Department of Genetics and Evolution, Mammal Research Institute Polish Academy of Sciences, 17-230 Białowieża, Poland; 2Museum and Institute of Zoology Polish Academy of Sciences, Nadwiślańska 108, 80-680 Gdańsk, Poland; s.czarnomska@gmail.com; 3Department of Chemistry and Bioscience, Aalborg University, Fredrik Bajers Vej 7K, 9220 Aalborg, Denmark; cp@bio.aau.dk; 4Aalborg Zoo, Mølleparkvej 63, 9000 Aalborg, Denmark

**Keywords:** ascertainment bias, single nucleotide polymorphism, Bialowieza Forest, reference pipeline, reduced representation sequencing, de novo pipeline, population genomics, STACKS, reference genome

## Abstract

**Simple Summary:**

The European bison is a dramatically low-diversified species, commonly analyzed using cattle-dedicated tools. Our aim was to compare two genotyping-by-sequencing (GBS) pipelines: de novo and reference pipeline, using the STACKS software and to reveal the maximum possible number of species-specific SNPs for our further project on European bison health. Therefore, we compared two genotyping-by-sequencing (GBS) pipelines: de novo (non-reference based) and a reference-based, using the STACKS software. We found a higher number of polymorphic loci from the reference pipeline in comparison to the de novo one. Next, we compared the results of the reference pipeline for the draft genome of European bison and completely annotated the *Bos taurus* genome. Higher numbers of polymorphic loci were revealed in European bison than in *Bos taurus* through the reference pipeline. We observed a possible effect of PCR duplicates on GBS data, as previously reported with the RADSeq approach. We recommend using a reference pipeline without PCR duplicates as a more efficient tool for species with low genetic diversity.

**Abstract:**

The European bison is a non-model organism; thus, most of its genetic and genomic analyses have been performed using cattle-specific resources, such as BovineSNP50 BeadChip or Illumina Bovine 800 K HD Bead Chip. The problem with non-specific tools is the potential loss of evolutionary diversified information (ascertainment bias) and species-specific markers. Here, we have used a genotyping-by-sequencing (GBS) approach for genotyping 256 samples from the European bison population in Bialowieza Forest (Poland) and performed an analysis using two integrated pipelines of the STACKS software: one is de novo (without reference genome) and the other is a reference pipeline (with reference genome). Moreover, we used a reference pipeline with two different genomes, i.e., *Bos taurus* and European bison. Genotyping by sequencing (GBS) is a useful tool for SNP genotyping in non-model organisms due to its cost effectiveness. Our results support GBS with a reference pipeline without PCR duplicates as a powerful approach for studying the population structure and genotyping data of non-model organisms. We found more polymorphic markers in the reference pipeline in comparison to the de novo pipeline. The decreased number of SNPs from the de novo pipeline could be due to the extremely low level of heterozygosity in European bison. It has been confirmed that all the de novo/*Bos taurus* and *Bos taurus* reference pipeline obtained SNPs were unique and not included in 800 K BovineHD BeadChip.

## 1. Introduction

The advent of massively parallel high-throughput sequencing (HTS) has dramatically altered the manner in which researchers conduct their research. This is certainly true for molecular population geneticists, who now consistently have access to large genetic datasets for non-model organisms. Genotyping by sequencing (GBS) is an efficient HTS technique. It involves next-generation sequencing (NGS) based on the reduced representation sequencing (RRS) strategy to obtain genome-wide single-nucleotide polymorphisms (SNPs). GBS provides an opportunity to discover polymorphic sites across individuals. It is a flexible, high-throughput assay capable of providing a sufficient marker density for genomic selection or genome-wide association studies in any population [1,2]. This technique has been widely used in plants for molecular marker discovery and genome-wide associations [3] and is an effective method in animals for SNP discovery and genotyping [4,5,6,7].

Moreover, GBS has been demonstrated as an effective method for genome-wide SNP discovery and genotyping, even in studies dealing with inbreeding control and genomic selection for example, in Indian cattle and Peking duck [8,9]. Several other studies using the GBS technique have been published, examining soybean [4], rice [10], oat [11], chicken [12,13], mouse [14], fox [15], and cattle [6], among others.

Genotyping by sequencing is one of the RRS approaches, where DNA adjacent to the restriction enzyme digestion sites is extracted, followed by NGS of the resultant fragments. The data obtained from GBS are then re-assembled into loci, anchored by the restriction enzyme digestion sites [16,17,18,19], and subsequently, variants are recognized and picked out across each locus. This technique is proven to be advantageous in reaching important regions of the genome that remain inaccessible by other sequence capturing approaches [2,20]. The sequencing approach based on restriction site-associated genomic DNA (i.e., RAD tags) was demonstrated by Baird et al. (2008) [17] for high-density SNP discovery and genotyping.

Previous genomic studies on the European bison (Bison bonasus) mainly utilized the genomic resources available for cattle. This approach turned out to be successful in a num- ber of studies [21,22,23,24,25,26,27,28,29] but also leaves doubt that the evolutionarily acquired changes of the species might have remained unknown due to the application of non-specific tools [30]. There is a dramatic demographic history of the species: extinction in the wild, recovery based on just seven founders with extremely unequal gene shares, and high inbreeding resulted in one of the lowest known genetic diversity in wild mammals. Therefore, we were interested in testing our approach for finding out the SNPs in a homozygous population. In this study, we aimed to estimate gains and losses of genomic information using different approaches of SNP acquisition: cattle-dedicated SNP tools and de novo and reference pipelines with and without PCR duplicates to avoid possibility of false alleles, as reported previously for RADSeq data [31,32]. The present study involves the optimization of a de novo pipeline, comparison of the de novo and reference pipelines, and comparison of the reference pipeline using draft genome and a fully sequenced genome integrated pipeline of STACKS on GBS data. Moreover, we have investigated the effect of PCR duplicates on our results. To our best knowledge, previous studies have compared the de novo (non-reference) and reference pipeline attempts of RADSeq data using STACK tool [18,33] but none has compared the analysis of de novo and reference pipelines using genotyping- by-sequencing (GBS) data, especially for low-variable, non-model species using STACKS software.

## 2. Materials and Methods

### 2.1. Sample Collection and Genomic DNA Extraction

Various soft tissues (muscles, heart, liver, and kidney) as well as blood samples from 256 European bison (211 males and 45 females), collected by the Mammal Research Institute, Polish Academy of Sciences in Bialowieza between 1990 and 2016, were used as DNA sources.

DNA extraction was performed using the following commercial total DNA isolation kits: Syngen DNA Mini Kit (spin-column protocol, Wrocław, Poland), Qiagen DNeasy^®^, Blood & Tissue Kit (spin-column protocol), and Sherlock AX Kit (Gdansk, Poland), A&A Biotechnology, a procedure with DNA precipitation, Gdansk, Poland), as per the manufacturer’s guidelines. Many of the materials available were blood samples, and the DNA was obtained using the phenol–chloroform extraction method with ammonium acetate [34].

### 2.2. Genotyping By-Sequencing Library Preparation and High-Throughput Sequencing

Genotyping-by-sequencing library preparation was performed following the proto- col elaborated by Elshire et al. (2011) [2] with the methylation-sensitive ApekI (4–5 cutter) restriction enzyme. High-throughput sequencing was performed on an Illumina HiSeq 4000 using 100 bp paired-end sequencing runs at BGI lab www.bgi.com) (accessed October 2019).

### 2.3. Processing of GBS RAD-Tags

Cleaning and demultiplexing of the raw sequencing data were performed using custom scripts at BGI lab. The sequencing output of the GBS generated a total of 1,393,551,064 raw reads. Downstream analysis was performed using STACKS version 2.41 (USA) [35] and VCFtools (U.K.) [36] for variant calling and filtering.

### 2.4. Bioinformatics Analyses

A quality check was performed on the cleaned and demultiplexed FASTQ files using the software FastQC 0.11.9 (https://www.bioinformatics.babraham.ac.uk/projects/fastqc/) (Cambridge, UK) [37]. FastQC reports containing read quality metrics were generated for all FASTQ files. We processed the sequenced data and analyzed reads from all samples using two integrated pipelines of the STACKS v.2.41 [35,38]. STACKS is a software that can handle reduced-representation-based GBS sequencing data with or without a reference genome. It can further identify SNPs as well as calculate population statistics. The software performs better with a high accuracy for SNP calling [39,40].

After cleaning and demultiplexing the raw data, construction of the STACKS catalogue, SNP calling, and genotype construction were performed for de novo and reference pipelines using the denovo_map.pl and ref_map.pl programs of the STACKS software.

For the reference pipeline, parameter optimization was not required. However, for the de novo pipeline, parameter optimization was a crucial step to obtain conclusive results from the analysis, particularly to facilitate the recovery of more loci for low-coverage datasets such as GBS [41,42]. To obtain orthologous loci, several key parameters were essentially optimized, as they affect the number of recovered polymorphisms [31]. We provided all command lines used in this work as supporting information (Appendix A).

#### 2.4.1. De Novo Pipeline

We ran a de novo pipeline using cleaned and demultiplexed fastq files. The STACKS de novo program follows several steps to obtain variants. By running the three STACKS components (*ustacks*, *cstacks*, and *sstacks*), alleles were identified from our population set as per the guidelines given in the STACKS manual [35,38].

A set of identical sequences is referred to as a “*stack*” and a putative locus is formed by merging several such “*stacks*”. The *ustacks* program aligned short read sequences into matching *stacks* from which loci are formed (putative alleles). Loci and SNPs were de- tected from matching *stacks* at each locus. A catalogue was created for all loci across all the samples with the *cstacks* program, using the optimized value of mismatches allowed between loci when building the catalogue. Furthermore, the *sstacks* program matched loci from each sample back to the catalogue. After completion of *ustacks*, *cstacks*, and *sstacks*, the tsv2bam program of the STACKS pipeline transposed the data and pulled in the set of paired-end reads that was linked with each assembled single-end locus.

There were four minimum parameters involved: “m” is for the minimum number of reads required for creating a stack/putative allele within individuals; “M” is the number of mismatches allowed between *stacks*/putative alleles within individuals to merge them into a putative locus; “n” is the number of mismatches allowed between *stacks*/putative loci between individuals during construction of the catalogue; and “r” (min. samples per pop) is the minimum percentage of individuals in a population required to process a locus for that population in particular [35,38,41,43]. We tested several combinations of parameter settings and also performed parameter optimization for this pipeline. The above com- ponents and workflow of the de novo pipeline are summarized in Figure 1 on the left side.

#### 2.4.2. De Novo Parameter Optimization

A total of 51 samples containing a maximum and minimum number of 6,333,140 and 3,054,686 reads, respectively, were used for de novo parameter optimization. The STACKS de novo program ran several times on each dataset, while varying the following parame- ters on each parse of program. We used the ‘M’ parameter (from *ustacks*) from 2 to 7 (M2–M7) and the ‘n’ parameter (from *cstacks*) from 2 to 7 (n2–n7). We consistently kept the value of ‘m’ = 3 and ‘r’ = 0.80, with the rest of the parameters on default setting. ‘r’ = 0.80 signifies that a locus has to be present in a minimum of 80% of individuals.

Therefore, the program first ran with ‘M’ = 2, ‘n’ = 2, ‘m’ = 3, and ‘r’ = 0.80, followed by ‘M’ = 3, ‘n’ = 3, ‘m’ = 3, and ‘r’ = 0.80, etc., up to ‘M’ = 7 and ‘n’ = 7. Furthermore, new polymorphic loci were identified across 80% of the population (r80 loci) for each incre- ment in parameters. We followed the procedure published by Paris et al. (2017) for pa- rameter optimization [18].

#### 2.4.3. Alignment and Variant Calling Using De Novo Pipeline (denovo_map.pl) of STACKS Software

Based on the parameter optimization described above, the values of M and n were kept equal to 4 (described in Results) with m = 3 and r = 0.80. These values were used for the whole dataset (256 samples). These parameters were run in the de novo pipeline, both with and without PCR duplicates.

#### 2.4.4. Alignment and Variant Calling Using Reference Pipeline (ref_map.pl) of STACKS Software Using *Bos taurus* and the European Bison Genomes

We used two different reference genomes: *Bos taurus* genome version UMD 3.1 [44] and the European bison genome [45]. Because the *Bos taurus* UMD 3.1 genome version was the source build of 800 K BovineHD BeadChip and has the complete chromosome information, while the European bison genome is available at the scaffold level. The draft genome assembly of the European bison was composed of 29,074 scaffolds with N50 of 4.7 Mb and 2.58 Gb size [45].

Burrows–Wheeler Aligner (BWA) version 0.7.17 [46,47] with the MEM algorithm was used to align all the quality-filtered sequencing reads, with default parameters, to *Bos taurus* and European bison genomes. Aligned files were converted from sequence alignment map (SAM) format to sorted, indexed binary alignment map (BAM) files using SAM tools 1.9 [48]. The “ref_map.pl” program was run with default parameters (model marukilow and var-alpha 0.05) to create a catalogue of SNPs across our sample set as a single popu- lation with and without PCR duplicates to process the sorted BAM files. The workflow of the reference pipeline is summarized in Figure 1 on the right side.

#### 2.4.5. Population Statistics

##### Measurements of Genetic Diversity

Several output format files generated by the integrated “population” program of STACKS. The “populations” program produced population summary statistics for both the de novo and reference pipelines. The population statistics file included the frequency of alleles, observed homozygosity, expected and observed heterozygosity, and fixation index (F_IS_) for every SNP found in a population as defined in the population map. The population haplotype statistics contained the frequency of haplotypes, genes, and haplotype diversity. The population genetics statistics summary contained population-level average summary statistics for all loci and variant (polymorphic) loci, including the average frequency of the major allele, average observed homozygosity, and average observed and expected heterozygosity in the populations. Following this, the “population” program further used to generate VCF (variant call format) files for each pipeline.

#### 2.4.6. Variant Filtering

After variant calling using de novo and reference pipelines, variant calling data (vcf format) obtained from the “population” program were further filtered to exclude variants with MAF < 0.05, max-missing 0.5, and minGQ 15, using VCFtools (0.1.16 (C)) [36]. We used the same stringency cut-offs across all data-processing pipelines (Table 1).

To calculate significance, we performed a chi-squared test for the H_E_ of the de novo and reference pipelines with and without PCR duplicates. (Table 2).

#### 2.4.7. Juxtaposition of SNPs Obtained from De Novo Approach, Reference Pipeline, and Bovine High-Density SNP Chip Tool

Each of the approaches generated a different number of SNPs, and to verify the hypothesis that we obtained the same markers using different pipelines, we decided to analyze their uniqueness in the de novo pipeline. Using the de novo pipeline with both reference pipelines for finding species-specific markers, final de novo assembled loci (catalog.fa.gz) obtained from the de novo pipeline were then aligned back to the reference genomes (*Bos taurus* and European bison genomes) using the BWA alignment tool [47]. Then, the stack_integrate_alignment script was used to integrate the alignment position for each locus back into the STACKS output files [18]. Then, the “population” program of STACKS ran again to obtain VCF files from each pipeline. Furthermore, we filtered the VCF files to exclude variants with MAF < 0.05, max-missing 0.5, and minGQ 15 using VCFtools (0.1.16 (C)) [36] (Table 3). We compared SNPs after filtering for *Bos taurus* reference (Table 1, SNP^2^) with de novo/*Bos taurus* (Table 2, SNP^2^) and European bison references (Table 1, SNP^2^) with de novo/European bison (Table 2, SNP^2^) using Linux commands (grep, awk) (Appendix A).

We also checked how many of our de novo/*Bos taurus* and *Bos taurus* reference obtained SNPs are species-specific and not doubled by the markers included in the 800 K BovineHD BeadChip, widely used in European bison studies. For that purpose, we compared the SNPs of the 800 K BovineHD BeadChip with markers obtained using de novo/*Bos taurus* and *Bos taurus* reference pipelines, based on their chromosome positions, using Linux commands (grep, awk) (Appendix A).

## 3. Results

### 3.1. Sequence and Variant Calling

The total number of raw sequences generated following high-throughput sequencing was 1,598,748,644 reads. Before demultiplexing, the number of reads per run varied from 48, 339,898 to 131,099,277, with an average of 88,819,369.1 and a median of 92,268,720 (Appendix A). The total number of reads after demultiplexing was 1,075,960,322, ranging from 295,738 to 6,333,140, with an average of 2,101,485.004 and a median of 1,949,360 (Appendix A).

### 3.2. De Novo Parameter Optimization Results

As described in the methods, we obtained the numbers of additional RAD polymorphic loci on each iteration of “M” and “n” from M2 to M7 and n2 to n7, respectively, keeping “r” = 0.80 and “m” = 3. Following this, we plotted the numbers of new polymorphic loci versus iterations of “M” in GNUPLOT (U.S.) (accessed on December 2020) (version 5.2, http://www.gnuplot.info/) and found that the number of polymorphic loci increased from M2/M3 to M4/M5, after which we observed a decrease (Figure 2). The highest number of r80 loci was obtained with M4/M5. Therefore, the value of “M = 4” at M4/M5 was chosen to perform de novo analysis for the entire dataset.

### 3.3. Comparison of Number of Catalogue Loci, Assembled Loci, Polymorphic Loci, and SNPs with and without PCR Duplicates Using De Novo Pipeline for European Bison Sequencing Data and Reference Pipeline for Bos taurus and European Bison

The de novo and reference pipelines were compared with and without PCR duplicates. The de novo pipeline was run for European bison sequencing data without any reference genome. The reference pipeline, on the other hand, was run while keeping both *Bos taurus* and European bison as reference genomes.

For each pipeline, we investigated the effect of PCR duplicates on (i) catalogue loci (ii) assembled loci, (iii) polymorphic loci, (iv) SNPs before filtering, and (v) filtered SNPs (using minGQ 15, max-missing 0.5, MAF 0.05; see methods for details). The numbers of catalogue loci, assembled loci, and polymorphic loci along with SNPs (before filtering) obtained using the reference pipeline (using *Bos taurus* genome and European Bison genome as a reference) were categorically higher than those obtained from the de novo pipeline. This trend remained consistent across values obtained with and without PCR duplicates. (Table 1)

Next, we compared the results from the reference pipeline for *Bos taurus* reference pipeline and European bison reference-based pipeline. We found that the number of catalogue loci was higher for *Bos taurus* reference pipeline with and without PCR duplicates, whereas the number of obtained assembled loci and polymorphic loci was higher for European bison reference pipeline. In addition, the number of SNPs obtained (both before and after filtering) was also higher for the European bison reference pipeline (Table 1).

Furthermore, we were interested in exploring the differences obtained when we included/excluded PCR duplicates in the reference pipeline and de novo pipelines. Based on the values obtained from both the reference pipeline and de novo pipelines, the catalogue loci, assembled loci, and polymorphic loci decreased when PCR duplicates were removed (Table 1). In addition, the total number of SNPs before filtering increased when PCR duplicated were not included in the de novo and reference pipeline, while the total number of SNPs after filtering increased when PCR duplicates were included in the denovo-based pipeline (Table 1). A similar trend was observed when we ran de novo pipeline/Bos taurus and de novo/European bison, with and without PCR duplicates (Table 2) (see the material method Section 2.4.7).

We plotted a comparison of the assembled loci (Figure 3A), polymorphic loci (Figure 3B), the total number of SNPs obtained before filtering (Figure 3C), and the total number of SNPs obtained after filtering (Figure 3D) with or without PCR duplicates, using the de novo and reference pipelines. The numbers of assembled and polymorphic loci were observed higher with PCR duplicates in comparison to without PCR duplicates using the de novo and reference pipelines (Figure 3A,B). The total number of SNPs before filtering were observed to be higher without PCR duplicates, using the de novo and reference pipelines (Figure 3C). From the trend obtained for filtered SNPs, we observed that the number of SNPs increased in the presence of PCR duplicates using the de novo pipeline, while it decreased with the reference pipeline (Figure 3D).

### 3.4. Population Statistics Results

As above, we performed a population statistics analysis on the results obtained from both the reference pipelines (performed on *Bos taurus* and European bison reference genomes) and the de novo pipeline (run for European bison sequencing data). Using the “population” program of STACKS (see methods), we deduced the percentage of polymorphic loci (%Poly.Loci) and the average observed and expected heterozygosity along with the average observed homozygosity for every locus, as well as several other parameters (Appendix A).

We observed a gain in %Poly.Loci upon removal of the PCR duplicates in both the reference and de novo pipelines. Keeping the PCR duplicates, the de novo pipeline showed higher expected as well as observed heterozygosity in comparison to the reference pipeline for all loci. However, interestingly, when PCR duplicates were removed, the de novo pipeline showed a lower expected heterozygosity and a higher observed heterozygosity than the reference pipelines for all loci (Table 3).

The value of average observed heterozygosity was almost the same in *Bos taurus* reference pipeline and European bison reference pipeline, when PCR duplicates were included in the reference pipelines for all and variant loci; these values showed a minor increase upon removal of the PCR duplicates for all loci. We observed a similar trend with the expected heterozygosity as well for all loci (Table 3).

For the above analysis, the chi-squared test was performed for Expected Heterozygosity for the de novo and reference pipelines. For the *Bos taurus* reference pipeline, the chi-squared statistic with Yates correction was X^2^ = 45.6594, *p* < 0.00001. For the European bison reference pipeline, the value of the chi-squared statistic with Yates correction was X^2^ = 51.1509, *p* < 0.00001. For the de novo pipeline, the chi-squared statistic with Yates correction was X ^2^ = 0.9804, *p* = 0.322102, non-significant; *p* > 0.05.

### 3.5. Juxtaposition of SNPs Obtained from de novo Pipeline, Reference Pipelines, and Bovine High-Density SNP Chip Tool

The results of the comparison of filtered VCF files from the de novo/*Bos taurus* with *Bos taurus* reference and de novo/European bison reference pipelines are presented in Table 4. Out of 67 SNPs (without PCR duplicates) obtained using the de novo/*Bos taurus* pipeline, only one was common in the de novo/*Bos taurus* and Bos taurus reference pipelines and 66 were unique in the de novo/*Bos taurus*. Out of 66 SNPs (without PCR duplicates) obtained using the de novo/European bison pipeline, only four were common in the de novo/European bison and European bison reference pipelines and 62 were unique in the de novo/European bison.

## 4. Discussion

This study compares de novo and reference pipelines to acquire the highest possible number of species-specific SNPs from a non-model organism. By comparing the results based on the analysis of both pipelines, we found more polymorphic markers in the reference pipeline as compared to the de novo pipeline. Our findings are in agreement with those of Torkamaneh et al. (2016), who performed a comparison of different SNP-calling pipelines on soybean (*Glycine max*) and discovered that four reference pipelines (TASSEL- GBS V1, IGST, TASSEL-GBSV2, and Fast-GBS) produced a higher number of SNPs than either of the two de novo pipelines (STACKS and UNEAK) tested did [40].

In addition, Shu M, Moran EV (2020) conducted a comparison between different SNP-calling pipelines on Ponderosa pine and found that two reference pipelines (TASSEL-GBS V2 and STACKS) identified more SNPs than either of two pipelines (UNEAK and STACKS) [49]. Shafer et al. (2017) [50] also suggested that by using a closely related reference genome in the RAD-Seq approach, the number of polymorphic loci can be increased, which is also supported by our results.

However, by contrast, Paris et al. (2017) [18] recovered a higher number of loci using de novo pipeline in comparison to a reference pipeline in RAD-Seq processing. In our case, we presume that the decreased number of SNPs from the de novo pipeline when compared to the reference pipeline might be explained by the extremely low level of heterozygosity of the European bison, which was reported in previous studies [24,27].

We also expected to obtain a higher number of SNPs from the de novo pipeline approach to enable a thorough genome-wide association study (GWAS) on the European bison. However, the results we obtained showed that using only the de novo pipeline approach in genomic analyses of a non-model species with low genetic variability might not be enough to achieve a sufficient number of SNPs for further use. Paris et al. (2017) [18] also explained that the assembled loci for each individual are matched to homologous loci to form a catalogue locus, and in the case of a de novo pipeline, sequences are aligned to each other. This could also explain the lower number of assembled and polymorphic loci, we obtained with the de novo pipeline analysis.

The number of SNPs after filtering obtained using both attempts (de novo and reference) is substantially lower than before filtering and when compared to cattle dedicated tools [24,25]. Nevertheless, they might have substantial potential in European bison studies, especially since this species has dramatically low diversity and the number of available SNPs is very limited [24,25]. As in RAD-Seq data analyses [18,31], our GBS-based results showed differences in the number of markers acquired when PCR duplicates were included in comparison to when they were removed (see Table 1 and Figure 2 in the results section). The presence of PCR duplicates increased the number of assembled loci and polymorphic loci in both pipelines. However, this effect could be misleading. As PCR duplicates may arise from multiple PCR products from the same template molecule binding on the flow cell, they can lead to false-positive variant calls [51]. Furthermore, our results showed that the presence of PCR duplicates decreased the values of expected heterozygosity (Table 3). Based on the results described above, we recommend removing PCR duplicates from GBS data, as also suggested previously for RADSeq data [31].

Table 3 shows the results of using three different approaches (de novo pipeline and reference pipelines for European bison and *Bos taurus*). The F_IS_ values close to zero indicate that the populations are in Hardy–Weinberg equilibrium. For each pipeline, we obtained comparable but dramatically low values of genetic diversity parameters, either with or without PCR duplicates. The expected and observed heterozygosity estimated in this study turned out to be far below the values reported previously for European bison [25,52]. We anticipate that these could be due to the use of cattle-dedicated, microsatellite, and SNP-chip-based tools.

We used the 800 K BovineHD BeadChip in this study, which we used in numerous previous projects. Over 99% of its markers were mapped to the UMD 3 bovine genome assembly (*Bos taurus*); thus we tested if any of the SNP markers we achieved using both methods (de novo/*Bos taurus* and *Bos taurus* references) had been included in the 800 K BovineHD BeadChip. It has been confirmed that all the de novo/*Bos taurus* and *Bos taurus* reference pipeline-obtained SNPs are unique. Their uniqueness is of importance as it means they have never been applied in previous GWAS studies on the European bison. We will thus be able to use them in future, expanded GWAS on susceptibility to one of the diseases that affects the species.

## 5. Conclusions

Our study shows that PCR duplicates in GBS analyses cause problems comparable to those in RADSeq processing. Their removal, as expected, enhanced the percentage of polymorphic sites detected and the values of expected heterozygosity.

Comparing the performance of de novo and reference STACKS pipelines for GBS data, we were intrigued to find that reference pipelines were able to detect a higher number of variants as compared to the de novo ones. Based on these results, for specific SNP detection using GBS processing, we recommend the use of specific genome assembly or a closely related reference genome in a less genetically variable species. If a de novo approach must be used, such as in the absence of a reference genome, we suggest using the standard STACKS as the most appropriate software for GBS.

In this project, the de novo attempt turnd out to be least effective as to the number of achieved SNPs, but all of them were species-specific and unique.

## Figures and Tables

**Figure 1 animals-11-02226-f001:**
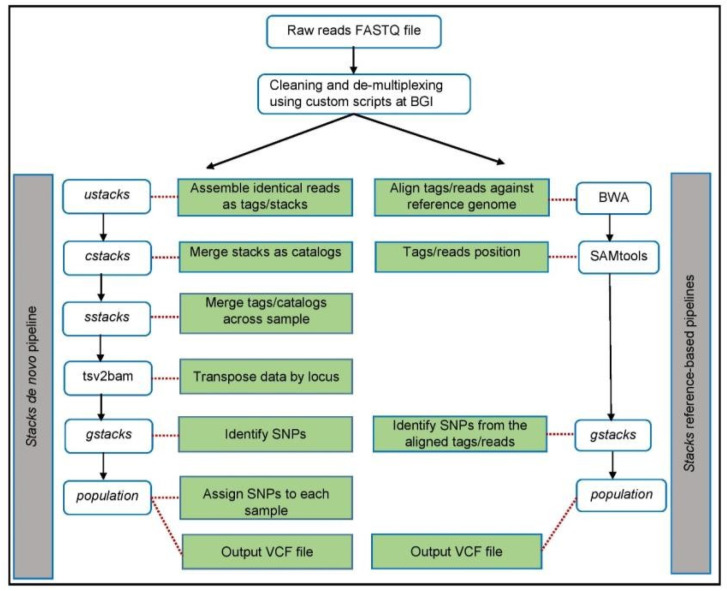
Comparison of de novo and reference pipelines. The horizontal boxes on the left side represent the programs in STACKS for a de novo pipeline. The horizontal boxes on the right side represent the programs in the STACKS for a reference pipeline. The green boxes in the middle represent potential program functions, while the dark red dotted lines specify the main function for each program in the two pipelines.

**Figure 2 animals-11-02226-f002:**
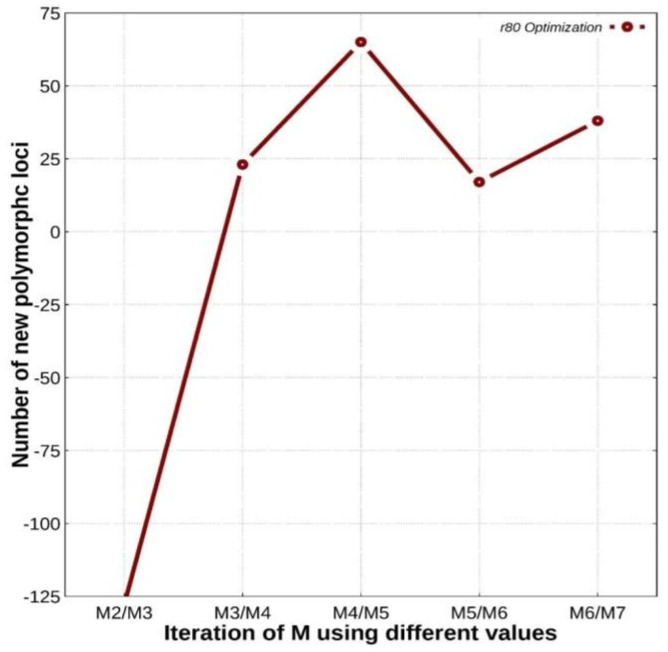
Plot of the number of new polymorphic loci (r80 loci) added on each iteration of “M” and “n” for datasets versus iterations of “M” using different values.

**Figure 3 animals-11-02226-f003:**
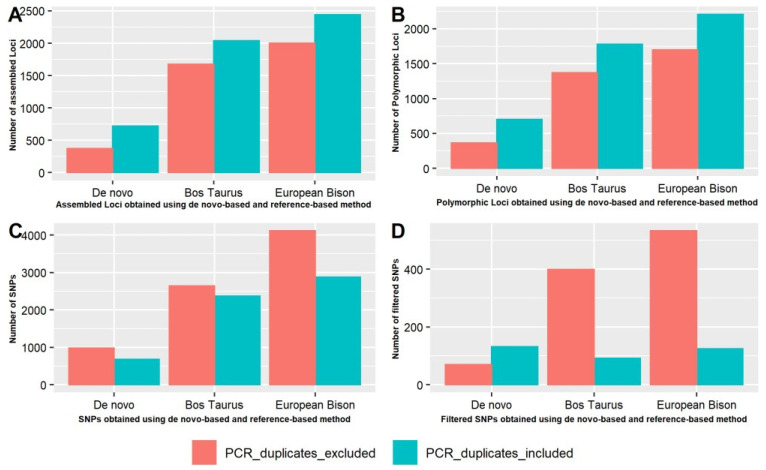
Effect of PCR duplicates on reference pipeline and de novo pipelines. Comparison of (**A**) assembled loci number, (**B**) polymorphic loci number, (**C**) SNP number obtained before filtering, and (**D**) filtered SNPs obtained before and after removing PCR duplicates.

**Table 1 animals-11-02226-t001:** Comparison of obtained numbers of catalogue loci, assembled loci, polymorphic loci, and SNPs using two different approaches: reference pipeline (for *Bos taurus (B. taurus)* and European bison (E. bison) and de novo pipelines, with and without PCR duplicates (PCR dupl. and no PCR dupl.)).

	*B. taurus* Reference:		E. Bison Reference:		De Novo	
	PCR dupl.	No PCR dupl.	PCR dupl.	No PCR dupl.	PCR dupl.	No PCR dupl.
Catalogue loci	6,811,878	6,417,724	6,597,114	6,158,207	2,874,175	2,812,858
Assembled loci	2036	1676	2441	2001	721	374
Poly. loci	1779	1370	2205	1697	699	362
SNPs ^1^	2371	2644	2878	4111	681	980
SNPs ^2^	92	399	124	532	132	70

Catalogue loci—number of genotyped loci; Assembled loci—number of assembled loci; Poly. loci—number of polymor- phic loci; SNPs^1^—SNPs before filtering; SNPs^2^—SNPs after filtering using minGQ 15, max-missing 0.5, and MAF 0.05.

**Table 2 animals-11-02226-t002:** Comparison of obtained numbers of catalogue loci, assembled loci, polymorphic loci, and SNPs using two de novo integrated approaches: (*Bos taurus* and European bison (E. bison) genome used as reference genome), with and without PCR duplicates (PCR dupl. and no PCR dupl.).

	De Novo/*B. taurus*		De Novo/E. Bison	
	PCR dupl.	No PCR dupl.	PCR dupl.	No PCR dupl.
Catalogue loci	2,817,544	2,756,852	2,823,558	2,762,878
Assembled loci	717	371	719	372
Poly. loci	695	359	697	360
SNPs ^1^	669	931	658	941
SNPs ^2^	122	67	111	66

Catalogue loci—number of genotyped loci; Assembled loci—number of assembled loci; Poly. loci—number of polymorphic loci; SNPs^1^—SNPs before filtering; SNPs^2^—SNPs after filtering using minGQ 15, max-missing 0.5, and MAF 0.05.

**Table 3 animals-11-02226-t003:** Population statistics calculated for all loci using different pipelines: reference pipeline (*Bos taurus* (*B. taurus*), European bison (E. bison)), and de novo, with and without PCR duplicates (PCR dupl. and no PCR dupl.).

	*B. taurus* Reference:		E. Bison Reference:		De Novo	
	PCR dupl.	No PCR dupl.	PCR dupl.	No PCR dupl.	PCR dupl.	No PCR dupl.
**All loci**						
%Poly.Loci	0.57431	0.87891	0.54228	1.06225	0.50444	1.39282
Obs_Het	0.00028	0.00047	0.00029	0.00048	0.00074	0.00061
Exp_Het	0.00017	0.00065	0.00017	0.00068	0.00047	0.00055
Fis	−0.00012	0.00174	−0.00012	0.00195	−0.00027	−0.00012
Var	0.00031	0.00226	0.00034	0.00241	0.00089	0.00029
StdErr	0.01999	0.022	0.01857	0.02042	0.04156	0.03731
**Variant loci**						
Obs_Het	0.04842	0.05308	0.0529	0.04529	0.14608	0.04369
Exp_Het	0.02969	0.0737	0.03146	0.06429	0.0922	0.03961
Fis	−0.02122	0.19808	−0.02303	0.18402	−0.05428	−0.00896
Var	0.05327	0.21865	0.06244	0.19384	0.17291	0.02092
StdErr	0.15438	0.20512	0.15109	0.15275	0.49407	0.28534

%Poly.Loci—percentage of polymorphic loci found within the population; Obs_Het—average observed heterozygosity per locus; Exp_Het—expected heterozygosity; FIS—average FIS estimations across loci.

**Table 4 animals-11-02226-t004:** Common and unique SNPs achieved using two different approaches: de novo and reference pipeline (using *Bos taurus* and European bison as a reference genome), with and without PCR duplicates.

		*B. taurus* Reference		E. Bison Reference	
		PCR dupl.	No PCR dupl.	PCR dupl.	No PCR dupl.
De novo SNPs	Common	16	1	22	4
	Unique	116	66	89	62

## Data Availability

The data presented in this study are available upon request from the corresponding author.

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
