# Peer review of "In Search of Species-Specific SNPs in a Non-Model Animal (European Bison (Bison bonasus))—Comparison of De Novo and Reference-Based Integrated Pipeline of STACKS Using Genotyping-by-Sequencing (GBS) Data"

_animals, 2021, doi:10.3390/ani11082226_

Round 1

Reviewer 1 Report

Even I have not accsses to the previous rounds. I noted that the paper has been intensely improved and it is now ready for its acceptance. Present manuscript has the sufficient quality to be published in Animals journal

Author Response

Thank you very much.

Reviewer 2 Report

This study performed SNP genotyping of European bison with two different approaches, de novo (non-reference based) and reference based pipelines. As a result, authors showed that the reference based approach obtained higher number of SNP loci than non-reference based one. Although I think that the obtained data in this manuscript is useful for publishing, the main object of this manuscript is ambiguous and more explanations are needed for introduction and discussion.

Major points:

1) In the simple summary, it is described that the aim of this study is to reveal the maximum possible number of species-specific SNPs. However, the reason why authors investigated it is unclear.

2) Authors compared non-reference based and reference based approaches, and concluded that reference based one is more useful. Such result has been fully investigated in previous studies. It is unclear why authors performed it.

3) Authors described that all SNPs obtained in these approaches were species-specific and unique. However, the most important thing for SNPs is their performance for future use (i.e., population study) rather than their uniqueness. Authors should discuss the advantages of SNPs obtained here.

4) Throughout the manuscript, explanation is not enough. For example, authors investigated the effect of PCR duplicates. However, authors did not explain why it needed.

Minor points:

Line 233, "grip" The linux command is "grep".

Line 358, "GWAS" Please add what it stands for.

Line 367, "might not seem satisfactory compared to ..." How many numbers of SNPs are enough? Please describe why authors seem so.

Round 2

Reviewer 2 Report

The manuscript has been revised well. I agree to accept this manuscript.